# Dynamics of spike-and nucleocapsid specific immunity during long-term follow-up and vaccination of SARS-CoV-2 convalescents

Nina Koerber [1,12], Alina Priller [2,12], Sarah Yazici [2,12], Tanja Bauer [1,3,12], Cho-Chin Cheng[1], Hrvoje Mijočević[1], Hannah Wintersteller [2], Samuel Jeske[1], Emanuel Vogel [1], Martin Feuerherd[1], Kathrin Tinnefeld[1], Christof Winter [4], Jürgen Ruland [4], Markus Gerhard [5], Bernhard Haller [6], Catharina Christa [1], Otto Zelger[7], Hedwig Roggendorf[2], Martin Halle[7], Johanna Erber[8], Paul Lingor[9], Oliver Keppler [10,3], Dietmar Zehn [11], Ulrike Protzer [1,3,12✉] & Percy A. Knolle [2,3,12✉]

Anti-viral immunity continuously declines over time after SARS-CoV-2 infection. Here, we characterize the dynamics of anti-viral immunity during long-term follow-up and after BNT162b2 mRNA-vaccination in convalescents after asymptomatic or mild SARS-CoV-2 infection. Virus-specific and virus-neutralizing antibody titers rapidly declined in convalescents over 9 months after infection, whereas virus-specific cytokine-producing polyfunctional T cells persisted, among which IL-2-producing T cells correlated with virus-neutralizing antibody titers. Among convalescents, 5% of individuals failed to mount long-lasting immunity after infection and showed a delayed response to vaccination compared to 1% of naïve vaccinees, but successfully responded to prime/boost vaccination. During the follow-up period, 8% of convalescents showed a selective increase in virus-neutralizing antibody titers without accompanying increased frequencies of circulating SARS-CoV-2-specific T cells. The same convalescents, however, responded to vaccination with simultaneous increase in antibody and T cell immunity revealing the strength of mRNA-vaccination to increase virus-specific immunity in convalescents.

[1] Institute of Virology, Helmholtz-Zentrum München/Technical University of Munich, School of Medicine, Munich, Germany. [2] Institute of Molecular Immunology and Experimental Oncology, Technical University of Munich, School of Medicine, Munich, Germany. [3] German Center for Infection Research (DZIF), Munich partner site, Munich, Germany. [4] Institute of Clinical Chemistry, Technical University of Munich, School of Medicine, Munich, Germany. [5] Institute of Medical Microbiology, Immunology and Hygiene, Technical University of Munich, School of Medicine, Munich, Germany. [6] Institute of Medical Informatics, Statistics and Epidemiology, Technical University of Munich, School of Medicine, Munich, Germany. [7] Institute of Sports Medicine, Technical University of Munich, School of Medicine, Munich, Germany. [8] Department of Internal Medicine II, Technical University of Munich, School of Medicine, Munich, Germany. [9] Department of Neurology, Technical University of Munich, School of Medicine, Munich, Germany. [10] Max von Pettenkofer Institute & Gene Center, Department of Virology, Ludwig-Maximilians-University Munich, Munich, Germany. [11] Institute of Animal Physiology and Immunology, Technical University of Munich, School of Life Sciences, Munich, Germany. [12] These authors contributed equally: Nina Koerber, Alina Priller, Sarah Yazici, Tanja Bauer, Ulrike Protzer, Percy A. Knolle. ✉email: protzer@tum.de; Percy.Knolle@tum.de

Severe acute respiratory syndrome coronavirus 2 (SARS-CoV-2) emerged in late 2019 and rapidly became pandemic[1–3]. SARS-CoV-2 infections cause a broad range of disease manifestations, and coronavirus-induced disease (COVID-19) already caused millions of deaths worldwide[4,5]. SARS-CoV-2 infects cells of the upper respiratory tract, but can also spread to other cells in the organism due to the widespread distribution of its receptor (angiotensin-converting enzyme 2-ACE2)[6]. SARS-CoV-2 specific B and T cell responses control infection in the majority of infected individuals[7–12], and virus-specific immunity has been shown to be generated after infection[13–21]. Particularly antibodies binding the receptor-binding domain of the SARS-CoV-2 spike protein are involved in protection from infection, but also T cells recognizing and eliminating infected cells contribute to control of infection[12,22–27]. Notwithstanding the important role of adaptive immunity, also innate immunity and particular interferon production is critical for efficient control of infection[28–30]. Auto-antibodies against interferons weaken the control of SARS-CoV-2 infection thereby posing a threat for developing COVID-19[31,32]. Strong local innate immunity in the upper airways further appears to be critical for rapid control of SARS-CoV-2 infection particularly in children[33,34]. On the other hand, systemic and overzealous immune activation after SARS-CoV-2 infection can cause immunopathology leading to severe COVID-19 (refs. [35–40]). Early and local containment of SARS-CoV-2 infection to prevent systemic and overzealous immune activation seems highly important to limit disease manifestation, which is confirmed by the success of COVID-19 vaccination establishing virus-specific immunity to prevent symptomatic disease in vaccinees[41–47]. These results support the notion that long-lasting memory immune responses directed against SARS-CoV-2 are important to prevent infection after vaccination or re-infection in convalescents.

Virus-specific T cell and antibody responses have been studied in individuals who recovered from SARS-CoV-2 infections and suffered only from mild symptoms as well as individuals who suffered from severe COVID-19[8,13,27,37,48–51]. It has remained unclear, however, how uniformly anti-viral immunity developed in convalescents from mild SARS-CoV-2 infection over time. Here, we report on the strength and dynamics of SARS-CoV-2 specific immunity in convalescents from mild SARS-CoV-2 infection during long-term follow-up before and after vaccination. While we observed a broad range of immune responses to infection, mRNA-vaccination uniformly boosted anti-viral immunity in all convalescents.

## Results

### The rapid decline of anti-SARS-CoV-2 IgG and virus-neutralizing antibodies in convalescents

We followed a cohort of seropositive individuals identified among 4554 health care workers at a university hospital from March 2020 onwards during the first and second wave of the pandemic and after vaccination. Information on the course of infection was obtained from each individual (Supplementary Fig. 1a–c). Ninety-one out of 4554 individuals tested positive by diagnostic polymerase chain reaction (PCR) for two SARS-CoV-2 genes and/or had seroconverted to anti-SARS-CoV-2 spike or nucleocapsid IgG, which was detected by at least two different commercial serological assays with >98% specificity each (assays from Abbott, Roche, YHLO, Mikrogen, and Euroimmune), for further details please see the section on study participants in "Materials and methods". This gave an overall specificity of at least 99.96% for the detection of previous SARS-CoV-2 infection. These 91 individuals were considered to have recovered from SARS-CoV-2 infection. All convalescents reported an uncomplicated course of infection, and 20 out of 91 convalescents reported no typical symptoms at all (Supplementary

Fig. 1c). Serum anti-SARS-CoV-2 anti-N antibody levels, that were detected by a chemiluminescence immunoassay (CLIA) in convalescents at month 2 after infection, had decreased significantly in most convalescents three months later (Fig. 1a–c). Accordingly, the virus-neutralization capacity of the convalescents' sera determined using a cell culture SARS-CoV-2 infection inhibition assay revealed a broad range of virus-neutralization activity, with nine convalescents lacking and twelve convalescents showing low virus-neutralization activity (Fig. 1d). Virus-neutralization activity decreased rapidly within 3 months (Fig. 1e), as also reported by other studies[13,49,52–55]. Of note, virus-neutralization activity but not anti-SARS-CoV-2 IgG levels correlated with reported typical symptoms of SARS-CoV-2 infection in convalescents, but not with sex, age or occupational exposure to COVID-19 patients, and anti-SARS-CoV-2 IgG levels correlated only weakly with virus-neutralization activity in convalescents (Fig. 1f, extended data Fig. 2a–f). Our results confirmed the rapid decline in anti-SARS-CoV-2 IgG and virus-neutralization activity in this cohort of health care workers after mild SARS-CoV-2 infection.

### Ex vivo detection of polyfunctional spike-reactive T cells by multicolor Fluorospot assays in convalescents of mild SARS-CoV-2 infection

Monitoring the numbers and functionality of T cells reacting to SARS-CoV-2 requires a quantitative and highly reproducible methodology, which is difficult to achieve by flow cytometric detection of SARS-CoV-2 reactive T cells. We have therefore applied an innovative multiplex Fluorospot-based assay that allows simultaneous detection of interferon-gamma (IFNγ)/interleukin 2 (IL-2)/tumor necrosis factor (TNF)/granzyme B (GzmB)-secretion in single T cells and can identify polyfunctional T cells, i.e., cells with simultaneous expression of different cytokines[56]. Combining the sensitivity and reproducibility of the ELISpot technology with multiparametric fluorescence-based analysis allowed us to determine the frequencies of SARS-CoV-2 spike-reactive (poly)functional cells directly ex vivo over time in a quantitative fashion and without in vitro expansion. Using a pool of overlapping peptides (15mers with 11aa overlap), that cover the entire SARS-CoV-2 spike antigen (peptide pool S1 covering aa1-643 and peptide pool S2 covering aa633–1273), we determined ex vivo the frequencies of spike-reactive cytokine-secreting cells as spot-forming cells (SFCs) per $10^6$ peripheral blood mononuclear cells (PBMCs). As a control, we stimulated PBMCs with a CMV/EBV/Flu (CEF) peptide pool that covers more than 95% of the most frequent HLA alleles[57]. This Fluorospot assay had high precision and robustness (Supplementary Fig. 3a, b), complying with the FDA guidelines for cell-based assays.

Employing the Fluorospot analysis for direct ex vivo characterization of PBMCs, we found higher frequencies of S1- or S2-reactive mono- and polyfunctional cytokine-secreting cells in convalescents of SARS-CoV-2 infection compared to naïve individuals (Fig. 2a–c, Supplementary Fig. 4a). Polyfunctional spike-reactive cells secreting ≥2 cytokines constituted approximately 30% of all cytokine-secreting cells (Fig. 2a, b), consistent with induction of functional and protective virus-specific T cell memory, which is characterized by polyfunctionality of T cells[56]. No increase in frequencies of spike-reactive GzmB-secreting cells was observed in convalescents ex vivo (Fig. 2a, b, d, Supplementary Fig. 4b), which pointed towards a low virus-specific CD8 T cell response. Of note, frequencies of spike-reactive TNF-secreting cells were high in convalescents but were also elevated in some naïve individuals, especially after S2 stimulation (Fig. 2a, b, d, Supplementary Fig. 4c), indicative of cross-reactivity for S2 peptides with seasonal coronaviruses[7,58,59]. In particular, high frequencies of IL-2-secreting cells (>140 cells/$10^6$ PBMCs) were

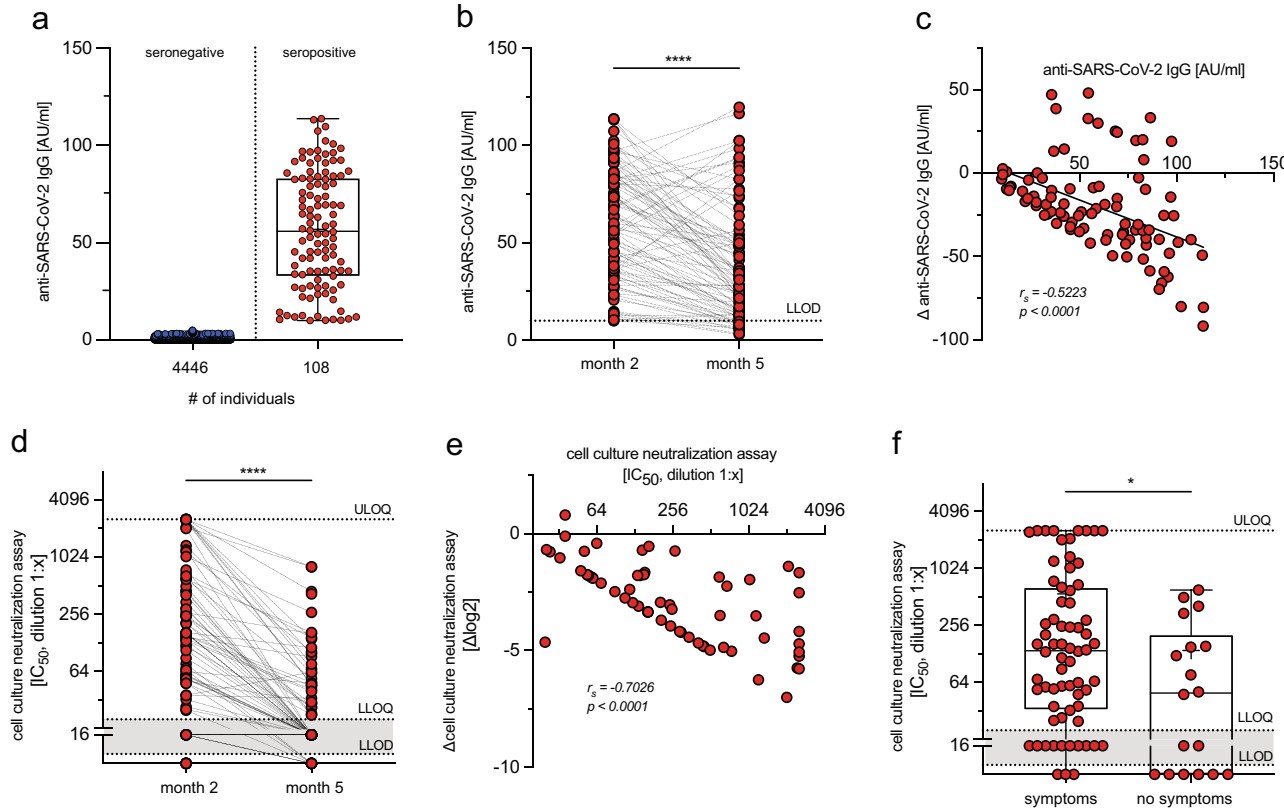

**Fig. 1 Dynamics of anti-SARS-CoV-2 IgG levels and virus-neutralization activity after mild SARS-CoV-2 infection. a** Anti-SARS-CoV-2 (nucleocapsid-specific) IgG levels determined by CLIA from 4554 health care workers (blue—seronegative individuals, $n = 4446$; red—seropositive individuals, $n = 108$) at month 2 after infection. Data are shown as median (55.93 and 0.40) with box bounds at 25% (33.02 and 0.26) and 75% percentile (83.04 and 0.69), whiskers show maxima (113.8 and 0.06) and minima (10.01 and 4.90). **b** Paired analysis of anti-SARS-CoV-2 nucleocapsid-specific IgG titers ($n = 91$). **c** Change (Δ) in anti-SARS-CoV-2 IgG titers at month 5 (y-axis) compared to initial anti-SARS-CoV-2 IgG titers at month 2 (x-axis); LLOD—lower limit of detection. **d** Paired analysis of virus-neutralization activity ($n = 86$) measured as 50% inhibition of viral infection in cell culture (dilution 1:x). Nine convalescents had no detectable and 12 had only low virus-neutralization activity. LLOQ lower limit of quantification, ULOQ upper limit of quantification. **e** Change (Δ) in cell culture virus-neutralization activity (as $\log_2$) at month 5 (y-axis) compared to virus-neutralization activity at month 2 (x-axis), results from 21 individuals were below the quantification limit and are not shown but included in the statistical analysis. **f** Reporting of symptoms ($n = 68$) or no symptoms ($n = 18$) in convalescents and virus-neutralization activity. Data are shown as median (138.70 and 49.45) with box bounds at 25% (33.40 and 1.00) and 75% percentile (634.70 and 201.30), whiskers show maxima (2560 and 604.40) and minima (1.00 for both). Statistical analyses by two-sided Wilcoxon signed-rank test (**b**, **d**), two-sided Mann–Whitney test (**f**), Spearman correlation and linear regression (**c**, **e**); $r_s$ denotes Spearman correlation coefficient; *$p < 0.05$; ****$p < 0.0001$.

detected in convalescents (Fig. 2a–d). Intracellular cytokine staining (ICS) and flow cytometric analysis detected cytokine expression preferentially by CD4 T cells (Fig. 2e, Supplementary Fig. 4d). In response to CEF peptide stimulation, which is known to be mediated by CD8 T cells[60], preferential IFNγ- and TNF- rather than IL-2-secreting cells were found in convalescent and naïve individuals (Fig. 2c, d, Supplementary Fig. 4a–c, e). Interestingly, frequencies of mono- and polyfunctional CEF-responses were higher in SARS-CoV-2-convalescents (Fig. 2c, d, Supplementary Fig. 4a–c, e), suggesting that SARS-CoV-2 infection either rendered antigen-specific T cells more responsive to stimulation or had increased their numbers. Frequencies of S1- and S2-reactive T cells directly correlated with frequencies of SARS-CoV-2 nucleocapsid-reactive T cells (Fig. 2f and Supplementary Fig. 4f), confirming the parallel development of spike and nucleocapsid-specific T cell immunity previously reported. Together, the sensitivity and robustness of the Fluorospot assay allowed us to directly quantify ex vivo the frequencies of spike-reactive polyfunctional T cells in convalescents after mild SARS-CoV-2 infection and detect a predominance of IL-2-secreting CD4 T cells.

Notwithstanding the predominant IL-2 profile, we also detected increased frequencies of spike-reactive interleukin 4 (IL-4) but

not interleukin 5 (IL-5) secreting cells in convalescents (Fig. 3a). However, Spearman's analysis revealed a direct correlation with anti-SARS-CoV-2 IgG levels and virus-neutralization activity only for S1- or S2-reactive IL-2- and IFNγ-secreting cells, while no correlation was found for numbers of TNF- or IL-4-secreting cells (Fig. 3b, c). As virus-neutralization activity rapidly declined, the correlation with IL-2 and IFNγ-secreting cells became weaker at later time points (Supplementary Fig. 5), pointing toward a limited value of virus-neutralization activity as a surrogate marker for cellular immunity. Notwithstanding the rapid reduction of virus-neutralization activity over time, our results showed the development of robust Th1-dominated immunity after mild SARS-CoV-2 infection, complementing earlier observations of discordant humoral and cellular immunity after infection[8,26,61].

The high inter-assay precision of the Fluorospot assay further allowed us to quantify spike-specific T cell responses over time. Compared to the rapid decline of SARS-CoV-2 specific antibodies and virus-neutralization activity, we found a less prominent but still significant reduction of the frequencies of spike-reactive IL-2- and IFNγ-secreting cells at month 11 after infection, whereas numbers of TNF-secreting cells remained unchanged (Fig. 4a, b). Of note, frequencies of CEF-reactive IFNγ and TNF-secreting

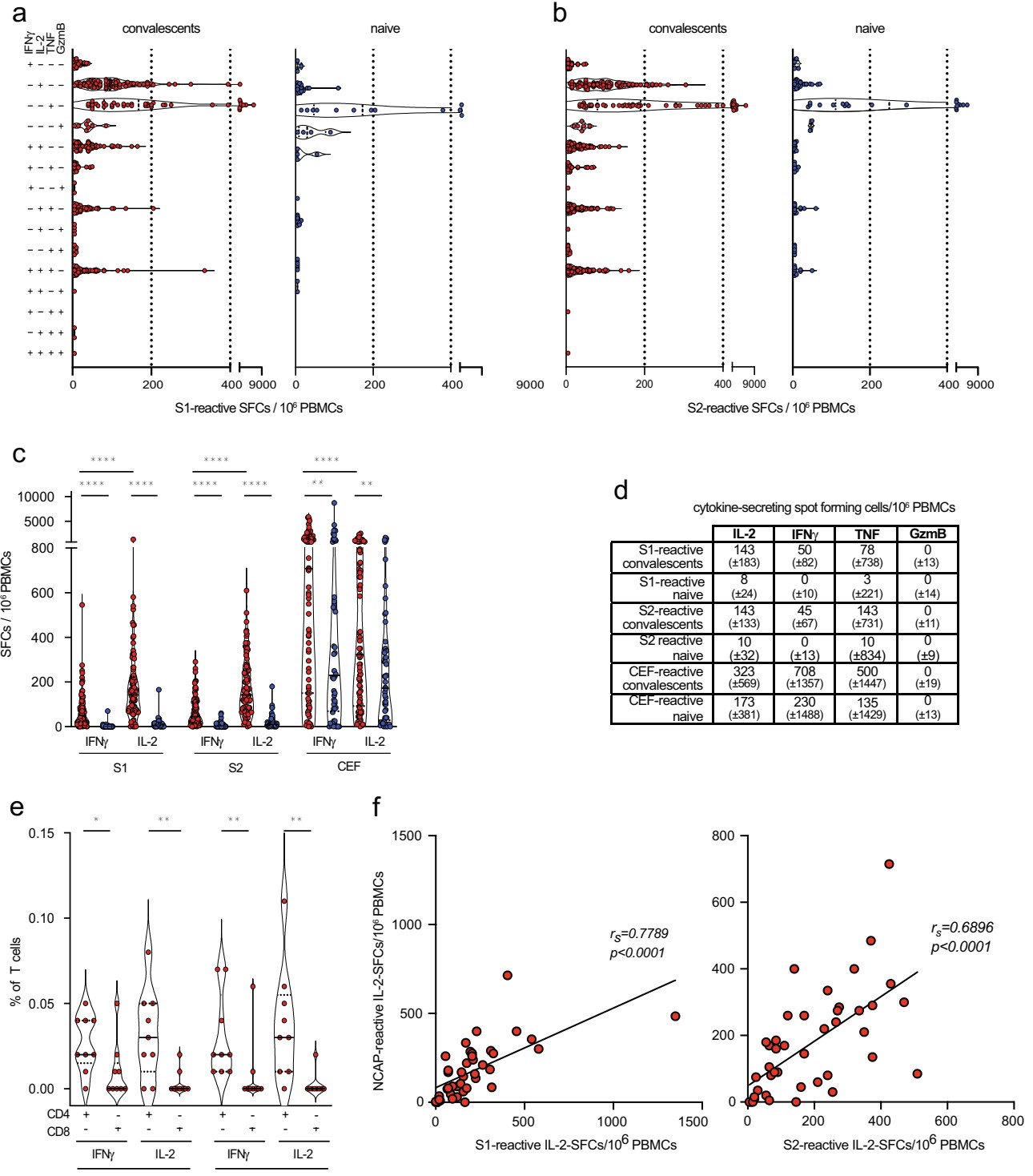

**Fig. 2 IL-2 dominated spike-specific CD4 T cell responses in convalescents. a, b** Ex vivo Fluorospot analysis revealing the frequencies of spike-reactive mono- and polyfunctional cytokine-secreting cells (spot-forming cells/SFCs per 10$^6$ PBMCs from convalescents (red) or naïve individuals (blue) at month 5. **c, d** Total frequencies of spike-reactive cytokine-expressing cells (shown as median ± SD), CEF (CMV/EBV/Flu) peptide pool as a control in convalescents (red, n = 88) or naïve individuals (blue, n = 52). **e** Ex vivo detection of the frequencies of CD4 and CD8 T cells of convalescents (n = 9) by intracellular cytokine staining and flow cytometry. **f** Correlation of the frequencies of nucleocapsid- (NCAP) and S1- and S2-reactive T cells determined by Fluorospot analysis in convalescents at month 5. Statistical analyses by two-sided Mann–Whitney, two-sided Wilcoxon signed-rank tests and Spearman correlation and linear regression (**c, e, f**); rs denotes Spearman correlation coefficient; *p < 0.05; **p < 0.01; ****p < 0.0001.

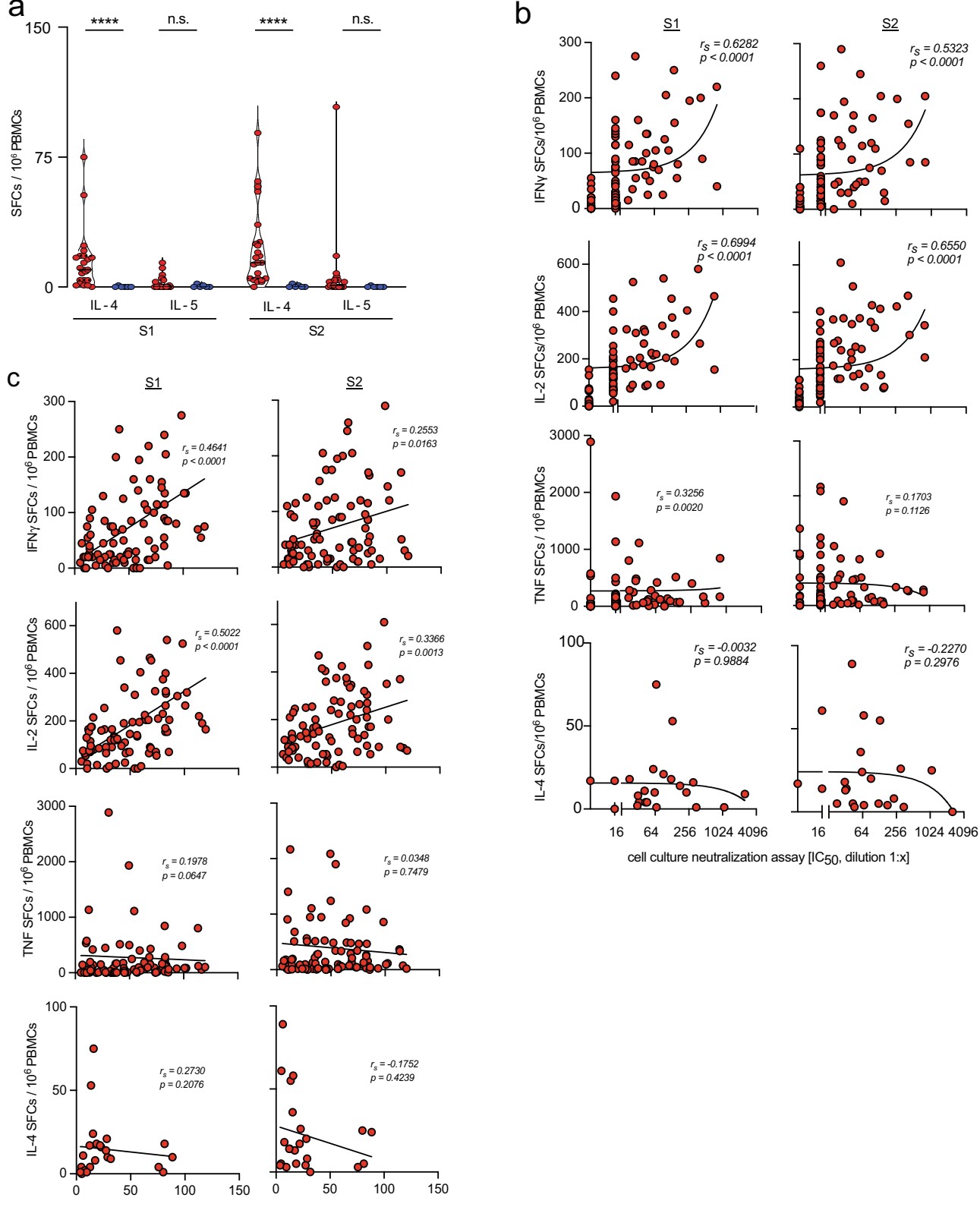

**Fig. 3 Frequencies of IL-2 and IFNγ-secreting but not IL-4 secreting cells correlate with virus-neutralization activity in SARS-CoV-2 convalescents.**
**a** Frequencies of spike-reactive IL-4- or IL-5-secreting cells in convalescents (red, $n = 23$) or naïve individuals (blue, $n = 6$). **b, c** Correlation of frequencies of S1- or S2-reactive IFNγ-, IL-2-, TNF-, or IL-4-secreting cells with virus-neutralization activity or anti-SARS-CoV-2 IgG in convalescents. Statistical analyses by two-sided Mann–Whitney test (**a**), Spearman correlation and linear regression (**b, c**); $r_s$ denotes Spearman correlation coefficient; n.s. denotes not significant; ****$p < 0.0001$.

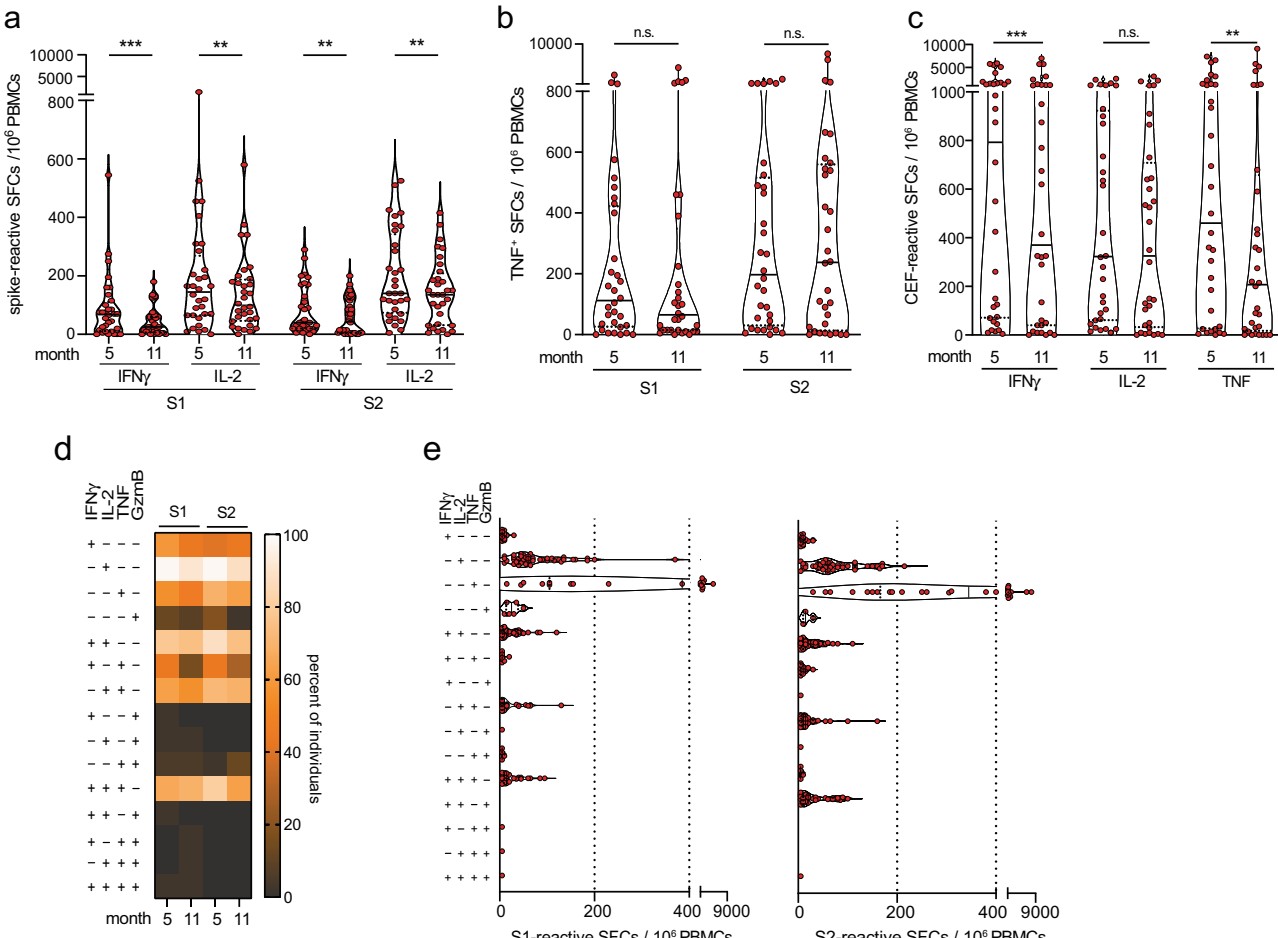

**Fig. 4 Persistence of polyfunctional S1- and S2-reactive IFNγ- and IL-2-secreting cells up to 11 months after SARS-CoV-2 infection. (a, b)** frequencies of spike-reactive cytokine-secreting cells in convalescents ($n = 32$) at month 5 and month 11 after SARS-CoV-2 infection. **c** Frequencies of CEF-reactive cytokine-secreting cells in convalescents ($n = 32$) at month 5 and month 11 after SARS-CoV-2 infection. **d** Heatmap revealing frequencies of convalescents bearing S1-reactive mono- or polyfunctional cytokine-secreting cells at month 5 and month 11 after SARS-CoV-2 infection. **e** Fluorospot analysis revealing the frequencies of spike-reactive mono- and polyfunctional cytokine-secreting cells (SFCs per $10^6$ PBMCs) from convalescents ($n = 49$) at month 11 after SARS-CoV-2 infection. Statistical analyses by two-sided Wilcoxon signed-rank tests (**a–c**, **e**); n.s. denotes not significant; **$p < 0.01$; ***$p < 0.001$.

cells declined from months 5 to 11 after SARS-CoV-2 infection (Fig. 4c). However, the proportion of T cells secreting two or more cytokines remained constant (Fig. 4d, e), consistent with sustained and functional virus-specific memory T cell immunity after mild SARS-CoV-2 infection[8,61–63].

**Rapid induction of polyfunctional IL-2-secreting spike-reactive T cells in convalescents and naïve individuals after BNT162b2 mRNA vaccination.** Since frequencies of IL-2 secreting CD4 T cells correlated with serum virus-neutralization activity (Fig. 3b, c), we quantified spike-specific T cell responses in convalescents ($n = 82$) compared to naïve individuals ($n = 53$) after prime and boost vaccination. In convalescents, we found a rapid increase of spike-reactive polyfunctional cytokine-secreting T cells already after prime vaccination, which did not further increase after boost vaccination (Fig. 5a–c, Supplementary Fig. 6a–c), consistent with previous reports[64–67]. In naïve individuals, we detected also a rapid increase in total and polyfunctional spike-reactive IL-2-secreting cells compared to IFNγ-secreting cells after prime vaccination (Fig. 5a–c, Supplementary Fig. 6a–c), which may support the production of virus-neutralizing antibodies after boost vaccination and further point towards induction of functional memory T cells[56]. Following boost vaccination, frequencies of spike-reactive cytokine-secreting cells showed no more

significant difference between convalescent and naïve individuals (Fig. 5a–c, Supplementary Fig. 6a–c), indicating a synchronization of spike-specific immune responses in convalescents and naïve individuals through BNT162b2 mRNA vaccination. According to a change in national vaccination guidelines during the follow-up period, convalescents continued to receive only a single vaccine shot. Thus, only 33/82 convalescents received both, prime and boost, BNT162b2 mRNA vaccinations. ICS revealed that after prime/boost vaccination mainly CD4 and to a lesser extent CD8 T cells produced IL-2, IFNγ or TNF in response to S1-/S2-stimulation, whereas after CEF-stimulation mainly IFNγ- and TNF-producing CD8 T cells were detected (Fig. 5d, e, Supplementary Fig. 6d, e). Of note, frequencies of spike-reactive IL-4-secreting cells determined by Fluorospot analysis were increased after prime/boost vaccination in convalescents and naïve individuals but surprisingly were higher in naïve individuals (Supplementary Fig. 6f). These results demonstrated a rapid and preferential induction of spike-reactive polyfunctional IL-2-secreting CD4 T cells in naïve individuals and convalescents already after one (prime) vaccination.

**SARS-CoV-2 convalescents lacking long-lasting virus-specific immunity respond to prime/boost BNT162b2 mRNA vaccination.** Next, surrogate virus-neutralization activity was

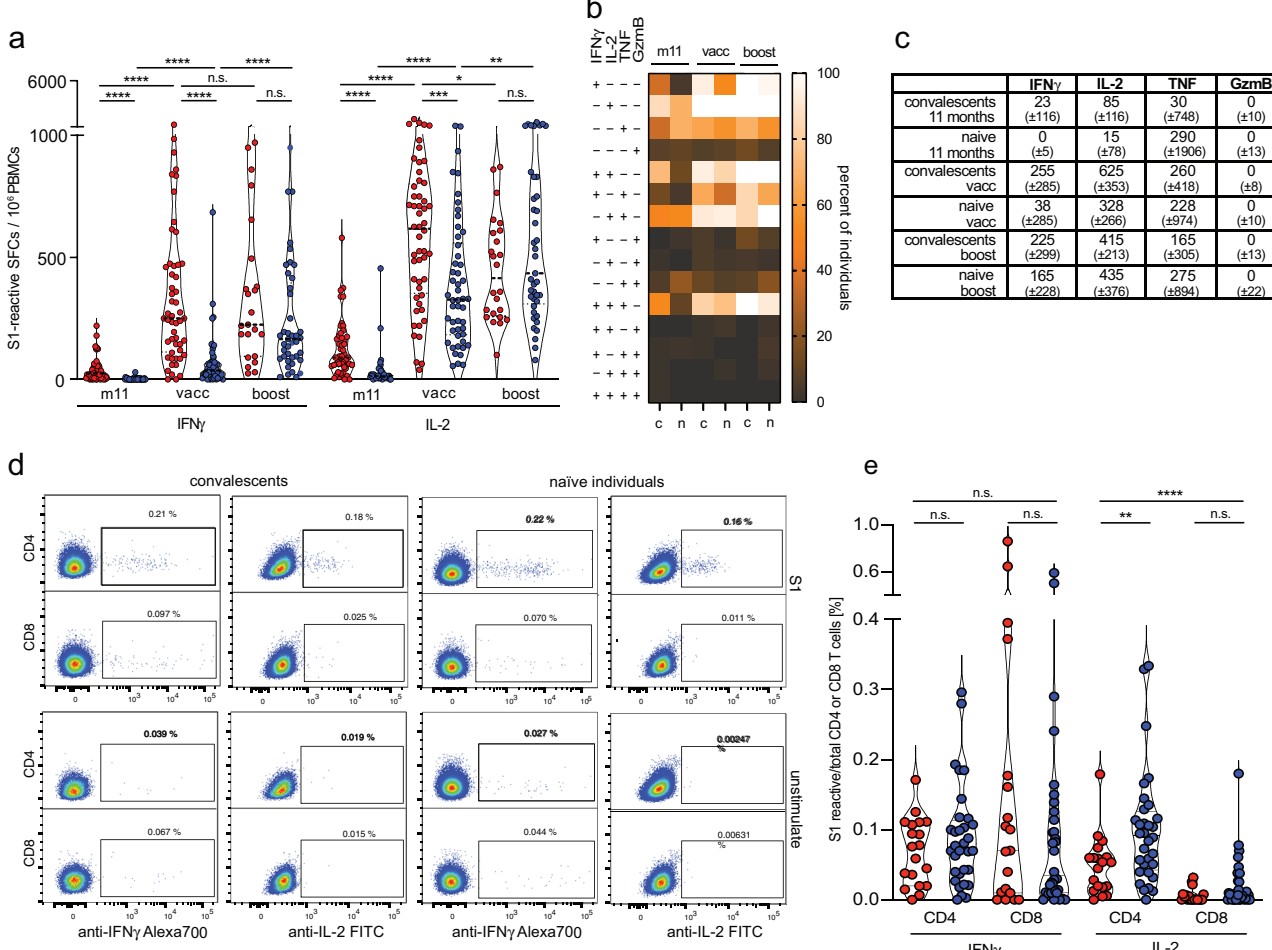

**Fig. 5 Rapid induction of polyfunctional IL-2-secreting T cells in convalescents and naïve individuals after BNT162b2 mRNA vaccination.**
(**a**) frequencies of spike-reactive cytokine-secreting cells in convalescents (red, n = 50 (m11), n = 54 (vacc), n = 23 (boost)), and naïve individuals (blue, n = 39 (m11), n = 49 (vacc), n = 40 (boost)) at month 11 after SARS-CoV-2 infection, two weeks after BNT162b2 mRNA prime vaccination (vacc) and 2 weeks after boost vaccination (boost). **b** Heatmap revealing frequencies of individuals bearing S1-reactive mono- or polyfunctional cytokine-secreting cells; convalescents (c), naïve individuals (n). **c** Median and standard deviation for S1-reactive cytokine-secreting cells. **d**, **e** S1-reactive IL-2 and IFNγ-producing CD4 and CD8 T cells in convalescents (red, n = 19) and naïve individuals (blue, n = 31) determined directly ex vivo by flow cytometry using intracellular cytokine staining. Statistical analyses by ANOVA, two-sided Mann–Whitney and two-sided Wilcoxon signed-rank tests (**a**, **e**). n.s. denotes not significant; *p < 0.05; **p < 0.01; ****p < 0.0001.

determined after BNT162b2 mRNA vaccination by a competitive CLIA. In convalescents, surrogate virus-neutralization activity reached maximal levels already after the first vaccination (2 weeks after prime vaccination 2099 ± 680; 2 weeks after boost vaccination 2022 ± 537 AU/ml), whereas in naïve individuals virus-neutralization activity remained low after the prime vaccination and required a boost vaccination for a sustained increase (2 weeks after prime vaccination 19 ± 110 AU/ml; 2 weeks after boost 1605 ± 717 AU/ml) (Fig. 6a), consistent with recent reports[47,64,65,68]. Since we detected a direct correlation between frequencies of IL-2-secreting cells and virus-neutralization activity in convalescents at month 5 after infection, it is likely that the early increase of IL-2-secreting CD4 T cells in naïve individuals after prime vaccination supports the production of virus-neutralizing antibodies after boost vaccination.

Surprisingly, in five SARS-CoV-2 convalescents (6.1%) we detected no increase in virus-neutralizing antibodies after BNT162b2 mRNA prime vaccination (Fig. 6a, Supplementary Fig. 7a), indicating a lack of spike-specific immunity in these convalescents. For comparison, in a separate cohort of 455 naïve individuals receiving a first BNT162b2 mRNA vaccination only

0.9% (4/455) showed comparably low levels of surrogate virus-neutralization activity (Fig. 6b, Supplementary Fig. 7b). Boost vaccination in these five low-responder convalescents, however, strongly augmented surrogate virus-neutralization activity, which then was similar to that observed in naïve individuals after boost vaccination (Fig. 6b). Since none of these convalescents reported immune-suppressive treatment or underlying disease, we suspected that these five convalescents did not generate long-lasting spike-specific immunity after infection.

The five convalescents originally presented with varying serum levels of antibodies against SARS-CoV-2 nucleocapsid and spike antigens (Supplementary Fig. 7c). These low-responder convalescents initially showed no virus-neutralization activity at all (Fig. 6c). They further had only low frequencies of spike-reactive and nucleocapsid-reactive cytokine-secreting cells, whereas the response to CEF peptide-stimulation was normal (Fig. 6d–g, Supplementary Fig. 7d), which excluded a general downregulation of T cell immunity in these convalescents. Of note, no polyfunctional spike- or nucleocapsid-reactive T cells were detected in these convalescents (Supplementary Fig. 7e). After BNT162b2 mRNA prime and boost vaccination four out of five

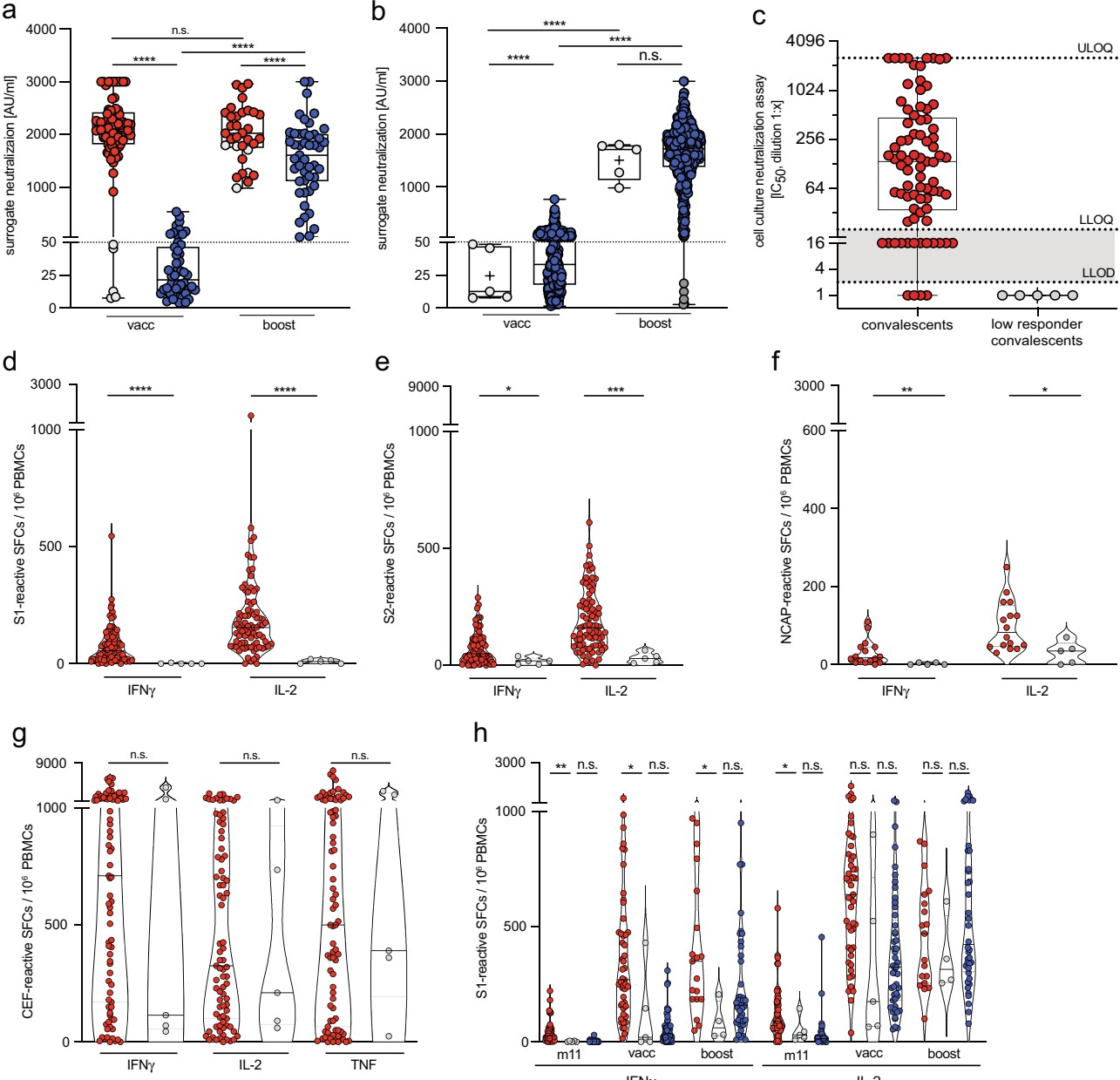

**Fig. 6 Low responder convalescents respond to prime/boost BNT162b2 mRNA vaccination.** (**a**) surrogate virus-neutralization after prime and boost vaccination in convalescents (red, $n = 77$ (vacc), $n = 28$ (boost)), low responder convalescents (gray, $n = 5$) and naïve individuals (blue, $n = 51$ (vacc), $n = 49$ (boost)). Data are shown as median (vacc: 2099 and 19, boost: 2022 and 1605) with box bounds at 25% (vacc: 1813 and 12, boost: 1742 and 1068) and 75% percentile (vacc: 2424 and 46, boost: 2414 and 2005), whiskers show maxima (vacc: 3000 and 531, boost: 2963 and 3000) and minima (vacc: 8 and 4, boost: 980 and 53). **b** Surrogate virus-neutralization of low responder convalescents (light gray, $n = 5$) and after prime and boost BNT162b2 mRNA vaccination compared to a cohort of 455 naïve individuals (blue) including low responders to vaccination (dark gray, $n = 4$). Data are shown as median (vacc: 13 and 33, boost: 1709 and 1723) with box bounds at 25% (vacc: 8 and 18, boost: 1126 and 1376) and 75% percentile (vacc: 47 and 68, boost: 1785 and 1981), whiskers show maxima (vacc: 48 and 764, boost: 1795 and 3000) and minima (vacc: 8 and 1, boost: 980 and 3). **c** Cell culture virus-neutralization activity at month 2 after SARS-CoV-2 infection in serum of convalescents ($n = 81$) compared to those with low response to vaccination ($n = 5$). Data is shown as median (136.4) with box bounds at 25% (34.2) and 75% percentile (479.5), whiskers show maximum (2560) and minimum (1.0). **d–g** Frequencies of spike-reactive, nucleocapsid-reactive or CEF-reactive cytokine-secreting cells in convalescents (red, $n = 83$ (**d**, **e**, **g**) and $n = 16$ (**f**)) and low responder convalescents (gray, $n = 5$) at month 5 after infection. **h** Frequencies of spike-reactive cytokine-secreting cells at month 11 after infection, 2 weeks after prime (vacc) and 2 weeks after boost (boost) BNT162b2 mRNA vaccination; convalescents (red, $n = 45$ (m11), $n = 49$ (vacc), $n = 19$ (boost)), low responder convalescents (gray, $n = 5$ at all time points) and naïve individuals (blue, $n = 39$ (m11), $n = 49$ (vacc), $n = 40$ (boost)). Statistical analyses by ANOVA, two-sided Mann–Whitney and two-sided Wilcoxon signed-rank tests (**a**, **b**, **d–h**). n.s. denotes not significant; *$p < 0.05$; **$p < 0.01$; ***$p < 0.001$; ****$p < 0.0001$.

low responder convalescents showed low frequencies of spike-reactive IL-2 and IFNγ-secreting cells that were similar to frequencies in naïve individuals rather than in convalescents after vaccination (Fig. 6h, Supplementary Fig. 7f, g), altogether corroborating a lack of virus-specific memory T cells. Of note, four of the low-responder convalescents did not report any symptoms during SARS-CoV-2 infection. Taken together, the response of low responder convalescents to prime/boost BNT162b2 mRNA vaccination, that was similar to naïve individuals, indicated that these individuals were in principle capable of mounting potent spike-specific immunity and suggested that particular circumstances during SARS-CoV-2 infection might have been responsible for the failure to establish long-lasting anti-viral immunity.

**The distinct magnitude of spike-specific immunity during follow-up of convalescents and after BNT162b2 mRNA vaccination.** During the second wave of the pandemic in fall and winter 2020/2021, i.e., between months 5 and 11 after the initial infection of our convalescents, and before BNT162b2 mRNA vaccination, we detected sharp increases in serum virus-neutralization activity by more than a factor of 8 (3 log$_2$) in 6 out of 82 convalescents, that was confirmed by a similar rise in the surrogate neutralization activity (Fig. 7a, b). Hereby, surrogate neutralization activity strongly correlated with cell-culture virus-neutralization activity (Supplementary Fig. 8a). No PCR-based diagnostic procedures were performed for the detection of SARS-CoV-2 RNA from nasopharyngeal swabs because none of the convalescents reported any symptoms. Surprisingly, no comparably steep rise in anti-nucleocapsid-specific antibodies was observed (Fig. 7c), indicating a selective surge of virus-neutralizing antibodies in these six individuals. Despite the sharp rise in virus-neutralization activity no increase in frequencies of spike-reactive or nucleocapsid-reactive cytokine-secreting mono- or polyfunctional T cells was detected by Fluorospot analysis (Fig. 7d–f, Supplementary Fig. 8b, c). It cannot be excluded, however, that T cell responses against non-structural SARS-CoV-2 antigens[48,69,70] increased but remained undetected in our study. In contrast, among the group of age- and sex-matched naïve control individuals ($n = 53$) only one person reported symptoms during that observation period and a PCR-based diagnostic procedure confirmed SARS-CoV-2 infection. In this individual, we found increased virus-neutralization activity and a simultaneous augmentation in anti-SARS-CoV-2 nucleocapsid-specific IgG as well as increased frequencies of spike-reactive IFNγ- and IL-2-secreting T cells (Fig. 7a–c, g, j, Supplementary Fig. 8d, e). This indicated that the disparate increase of virus-specific antibody titers but not T cells observed in the six convalescents was unlikely to result from productive re-infection.

COVID-19 vaccination offered us the unique opportunity to compare the magnitude of spike-specific immunity after challenge. After vaccination, convalescents with a prior increase in virus-neutralization activity showed both, a further increase in virus-neutralization activity and increased frequencies of spike-specific IFNγ- and IL-2-secreting T cells, that were not distinct from those of the other convalescents (Fig. 7i, Supplementary Fig. 8f–i). Also, the previously infected individual showed a further increase in numbers of IFNγ- and IL-2-secreting cells after BNT162b2 mRNA vaccination (Fig. 7j, Supplementary Fig. 8j, k), demonstrating the robust strength of vaccination to boost virus-specific B and T cell immunity in convalescents.

## Discussion
We here show in SARS-CoV-2 convalescents with asymptomatic or mild SARS-CoV-2 infection the persistence of low frequencies of polyfunctional spike-reactive T cells over more than 11 months after infection, with polyfunctional IL-2 producing CD4 T cells being preferentially present and correlating with SARS-CoV-2-neutralizing antibody levels. Using a multicolor Fluorospot assay to detect the simultaneous release of cytokines in spike-specific T cells directly ex vivo, we add further evidence for the sustained presence of polyfunctional cytokine-producing memory T cells after mild SARS-CoV-2 infection[8,16,21,67,71].

However, we found that 6% of convalescents did not develop virus-specific memory T cell responses and lacked virus-neutralizing antibodies after SARS-CoV-2 infection, and responded to vaccination with a similar dynamic as naïve individuals. Our testing strategy for detection of resolved SARS-CoV-2 infection in health care workers included PCR-based diagnostics and/or use of at least two different serological assays, which resulted in an overall specificity of at least 99,96% for defining SARS-CoV-2 convalescents. Failure to mount a memory response to vaccination in convalescents was associated with low virus-neutralizing antibody titers, low frequencies, or absence of spike as well as nucleocapsid-reactive T cells early after SARS-CoV-2 infection. We cannot exclude, however, that specific immune responses against other parts of SARS-CoV-2, such as non-structural antigens[48,69], might have developed that were not investigated here. This lack of induction of long-lasting adaptive immunity might have resulted either from infection with too few viruses or contact with the poorly infectious virus, where innate immunity in the upper respiratory tract might have sufficed to achieve early control of infection[28,34] but may have failed to induce lasting adaptive immunity. Alternatively, also genetic restraints may have caused individuals low SARS-CoV-2 specific immunity[72].

The rapid decline of virus-specific antibodies, reported to occur after mild SARS-CoV-2 infection[73–75], was also observed in our cohort and led to a drop of virus-specific antibodies below the detection limit. Since these convalescents responded to COVID-19 vaccination with induction of spike-specific B and T cell immunity, we can exclude a complete non-responsiveness to SARS-CoV-2 antigens. It is noteworthy that prime and boost vaccinations were necessary for these low-responder convalescents to mount spike-specific immunity. The current recommendation to vaccinate SARS-CoV-2 convalescents only once may therefore leave some convalescents with incomplete protection from further infection.

While virus-neutralizing antibody levels continuously declined in all convalescents over time, 8% of convalescents showed a sudden increase of virus-neutralizing antibody titers during the second wave of the pandemic. Persistent SARS-CoV-2 infection is observed in immunosuppressed individuals[76–78] and is rather unlikely to be the reason for the observed sudden rise in virus-neutralizing antibodies in healthy convalescents. However, the establishment of latent infection has recently been proposed for RNA viruses such as Ebola virus[79], and may potentially also apply for SARS-CoV-2 that can infect immune-privileged sites such as the central nervous system[80]. Furthermore, it is possible that an increase in antibody affinity after SARS-CoV-infection[54] may have contributed to the observed increase in virus-neutralization activity in convalescents. Interestingly, the increase in virus-neutralization activity was neither accompanied by increased levels of nucleocapsid-specific antibodies nor by increased frequencies of circulating spike or nucleocapsid-specific T cells. We cannot exclude, however, an increase in frequencies of T cells directed against nonstructural SARS-CoV-2 antigens, that were not investigated here. Alternatively, abortive infection with SARS-CoV-2 or contact with the poorly infectious virus may have caused the increase in virus-neutralizing antibody titers. In such a scenario, a locally restricted immune response in the upper respiratory tract and/or the lung against SARS-CoV-2 may have

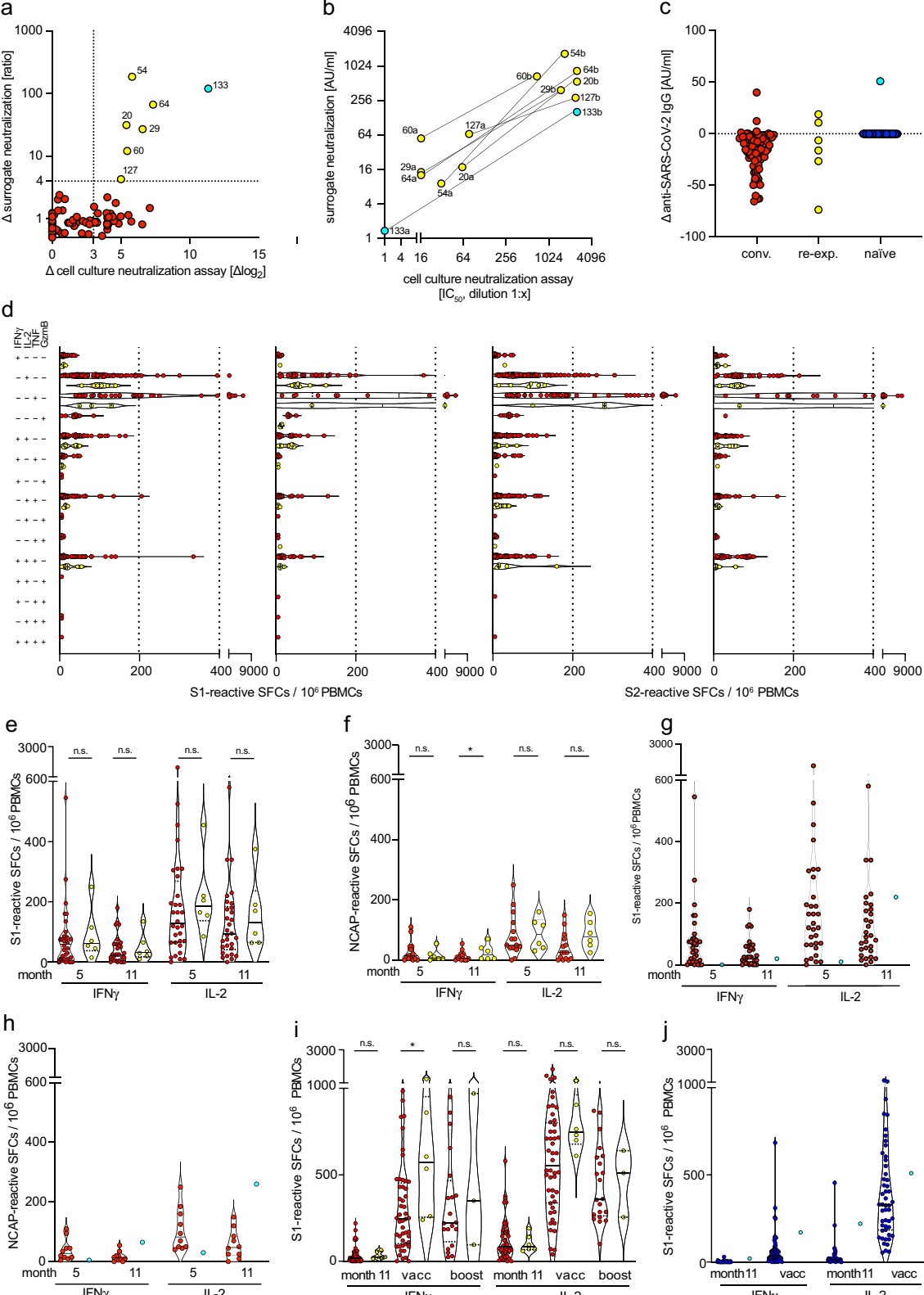

developed. In fact, memory B cells reside in the lung after influenza infection[81], and resident spike-specific memory B cells in the lung or upper respiratory tract of SARS-CoV-2 convalescents may be activated through contact with SARS-CoV-2. Likewise, T cells resident in the lung or the upper respiratory tract locally control respiratory tract virus infection[82–84], and SARS-CoV-2 specific T cells residing in the lung of convalescents[85] may

contribute to control of infection. Such local control of SARS-CoV-2 infection by virus-specific immune cells resident to the upper respiratory tract would escape our immune monitoring strategy, which can only evaluate circulating immune cells. However, immune cells act in tissues but not the circulation[86], and virus-specific immunity at the site of infection may be an efficient way to control infection and prevent the spread of

**Fig. 7 Disparate increase in virus-neutralization activity but not spike-reactive T cells in convalescents during long-term follow-up compared to infection and vaccination. a** Change ($\Delta$) in cell culture virus-neutralization activity and surrogate virus-neutralization activity in convalescents (red, $n = 78$), and convalescents with more than 3 $\log_2$ change in cell culture neutralization activity (yellow, $n = 6$) (detected in samples obtained at months 5, 8, and 11) or in a recently SARS-CoV-2 infected naïve individual (1/53) (turquoise, $n = 1$). **b** Virus-neutralization activities at two-time points in convalescents identified in (**a**); a —denotes values at month 5 or 8, and b—denotes values at month 11 after initial infection. **c** Change ($\Delta$) in nucleocapsid-specific anti-SARS-CoV-2 IgG levels in individuals from (**a**) and naïve individuals (blue, $n = 52$). **d** Frequencies of S1/S2-reactive mono- and polyfunctional cytokine-secreting cells in convalescents from (**a**, yellow) at month 5 (left panel) and at month 11 (right panel) as well as convalescents without a surge in virus-neutralization activity (red). **e** Total frequencies of S1-reactive IFNγ- and IL-2-secreting cells from (**d**). **e, f** Frequencies of spike-reactive or nucleocapsid-reactive cytokine-secreting cells in convalescents (red, $n = 32$ (month 5), $n = 30$ (month 11)) or convalescents with increased virus-neutralization activity (yellow, $n = 6$). **g, h** frequencies of spike- and nucleocapsid-reactive cytokine-secreting cells in a recently infected naive individual from (**a**, turquoise) and convalescents (red), at months 5 and 11 (after infection of the single individual). **i, j** Frequencies of S1-reactive IFNγ- and IL-2-secreting cells before vaccination (month 11), 2 weeks after vaccination (vacc) and 2 weeks after boost vaccination (boost) in convalescents (red, $n = 44$ (month 11), $n = 48$ (vacc), $n = 20$ (boost)) and convalescents with increased virus-neutralization activity (yellow, $n = 6$ at all time points), and in naïve individuals (blue, $n = 39$ (month 11), $n = 49$ (vacc), $n = 40$ (boost)), and an infected individual (turquoise, $n = 1$). Statistical analyses by ANOVA, two-sided Mann–Whitney and two-sided Wilcoxon signed-rank tests (**e, f, i**). n.s. denotes not significant; *$p < 0.05$.

infection to other organs. The local presence of virus-specific memory immune cells in the upper respiratory tract may also provide an explanation for the distinct additional protection from re-infection observed for convalescents[87]. Of note, breakthrough infections in vaccinated or convalescent individuals, that likely would result in a simultaneous increase of virus-specific antibodies and T cells, have mainly been reported for the highly infectious alpha and delta SARS-CoV-2 variants[88–91], neither of which was present during the second wave of the pandemic in Germany. Interestingly, after mRNA vaccination of the same individuals, we detected concordant increases in circulating virus-neutralizing antibodies and spike-specific T cells, which suggests a prominent boosting effect of vaccination in all convalescents. The increased protective function of virus-neutralizing antibodies detected in convalescents after COVID-19 vaccination[19] may further complement tissue-resident virus-specific immune cells to successfully contain infection with SARS-CoV-2 or the more infectious alpha and delta variants. Taken together, our results advocate the use of vaccination in individuals with prior mild SARS-CoV-2 infection to further enhance immune protection for the prevention of re-infection.

## Methods

**Study participants in seropositive and seronegative cohorts.** In April/May 2020, 4554 employees of the University Hospital München Rechts der Isar of the Technical University of Munich underwent SARS-CoV-2 IgG testing after the first wave of the SARS-CoV-2 pandemic. Seropositive and a matched group of seronegative individuals, who gave written informed consent, were included in a prospective follow-up study (SeCoMRI) that was extended for further follow-up after BNT162b2 mRNA vaccination (VaCoMRI). Both studies were approved by the local ethics committee of the Technical University of Munich (ethics vote 476/the 20S and 26/21S-SR).

We identified 108 individuals who tested positive for anti-SARS-CoV-2 IgG by an indirect chemiluminescence immunoassay (iFlash CLIA, YHLO Biotechnology, China; specificity 99.3% (98.3–99.7), sensitivity 94% (89.0–96.8))[92,93], among whom 94 individuals agreed to participate in the follow-up study. Among these, positive anti-SARS-CoV-2 IgG testing was confirmed in 89 participants by at least two further commercial assay systems, and in one further assay system in two individuals, for which material was limited, resulting in a combined specificity of at least 99.94%. Confirmatory SARS-CoV-2 IgG tests were performed in the following order: Roche Elecsys (specificity 99.1% (96.7–99.7), sensitivity 92.7% (87.3–95.9))[92,94]; Abbott Architect (specificity 99.5% (98.5–99.8), sensitivity 90.0% (84.2–93.8))[92,95]; Mikrogen Diagnostics (Germany) recomLine blot (specificity 99.7%, sensitivity 95.2–100% according to the manufacturer); Euroimmune (Germany) anti-S1 IgG ELISA (specificity 99.2% (98.1–99.6), sensitivity 78.0% (70.7–83.9))[92]. Three individuals only tested positive for anti-SARS-CoV-2 IgG using the screening assay (Yhlo, iFlash), and were therefore excluded from the study.

In 34/91 seropositive individuals with at least two independent positive anti-SARS-CoV-2 IgG tests, the infection had been confirmed by PCR-testing for SARS-CoV-2 RNA (for details see supplementary Table I). Thus, 91 convalescents of mild SARS-CoV-2 infection were identified and included in the follow-up study. The

specific characteristics of this cohort of convalescents are shown in Supplementary Fig. 1c. A control cohort of naïve individuals for follow-up studies was randomly selected from the 4446 seronegative individuals by matching for sex, age, working conditions, and risk factors that were present in the seropositive cohort (Supplementary Fig. 1c). After follow-up for 9 months, study participants were offered to continue to follow up after BNT162b2 vaccination (VaCoMRI). A total of 82 convalescents and 51 naïve individuals gave informed written consent to participate in the follow-up study. The first 23 convalescents received prime and boost BNT162b2 mRNA vaccination 28 days apart. Due to a change in the national guidelines remaining convalescents received only a single shot of BNT162b2.

**Serum SARS-CoV-2 neutralization activity.** Serum neutralization activity was determined using an in-house cell culture-based infection inhibition assay and a surrogate neutralization assay based on the competition with binding of a labeled recombinant angiotensin-converting enzyme-2 to microparticle-coupled recombinant SARS-CoV-2 receptor-binding domain (SARS-CoV-2 NAb, YHLO Biotechnology) used according to the manufacturer's instruction on the automated iFlash 1800 platform.

For infection inhibition, SARS-CoV-2 (D614G variant of the Wuhan strain) isolated from a nasal swab sample of one of the first patients in Germany in February 2020 was grown on VeroE6 cells. The full-length viral genome sequence has been uploaded into the GISAID database, accession ID: EPI_ISL_582134. For the infection inhibition assay, VeroE6 cells (ATCC, US) were seeded in supplemented Dulbecco's Modified Eagles medium (Thermo Fisher Scientific, Germany) at 15,000 cells per well in a 96-well plate one day before infection was started using SAR-CoV-2 at a multiplicity of infection of 0.06 plaque-forming units (PFU)/cell. To detect virus-neutralization activity, serum samples were serially diluted 1:2 with DMEM up to a 2048 dilution and added to SARS-CoV-2 (900 PFU/15,000 cells/well) in a total volume of 50 μL at 37 °C. After one hour of preincubation, the inoculum was transferred to the pre-seeded VeroE6 cells for another one-hour incubation at 37 °C before the inoculum was replaced by supplemented DMEM. SARS-CoV-2 infection was terminated after 24 h by adding 4% paraformaldehyde to fix the cells, and the infection rate was analyzed by an in-cell ELISA. After fixation, cells were washed with PBS and permeabilized with 0.5% saponin (Sigma-Aldrich, Germany). Blocking buffer consisting of 0.1% saponin–10% goat serum (Sigma-Aldrich) in PBS was added and incubated for one hour on fixed cells to avoid unspecific binding of antibodies. Afterward, viral double-stranded (ds) RNA inside cells were detected by anti-dsRNA J2 monoclonal antibody (Jena Bioscience, #RNT-SCI-10010500, Germany; diluted 1:500) overnight incubation at 4 °C and followed by 1-h incubation under room temperature with secondary antibody goat anti-mouse IgG2a-HRP (Southern-Biotech, USA; 1:2000 diluted). Washing steps with 0.01% Tween20–PBS buffer were included in between and after. Finally, signals were created by the interaction between HRP and the substrate, and the reaction was stopped by the addition of 2 N sulfuric acid solution, continued by optical detection with a Tecan Infinite 200 reader (TECAN, Switzerland) at 450 nm wavelength. The inhibition curve of each sample was analyzed by statistical analysis software Graph Pad Prism (GraphPad Software, USA), and 50% inhibitory concentration (IC$_{50}$) was determined using non-linear regression.

**Isolation and cryopreservation of peripheral blood mononuclear cells (PBMCs).** Blood from study participants was drawn with the Vacutainer CPT™ System into sodium citrate CPT tubes (Becton Dickinson Biosciences, USA) and tubes were mixed five times before storing them upright at room temperature. Within two hours of blood collection, CPT tubes were centrifuged in a horizontal rotor (swing-out head) (1800$g$, 15 min, RT). Next, plasma was removed and PBMCs were transferred to 15 mL polystyrene Falcon tubes and mixed with 10 mL

PBS by gently inverting the tubes five times. PBMCs were centrifuged (300 g, 10 min, RT) twice in 10 ml of PBS. For counting, cells were resuspended in CTL Test medium (CTL Europe GmbH, Germany) and 10 µL of the cell suspension was diluted 1:2 with CTL Live/Dead cell counting dye (CTL-LDC Live/Dead cell counting kit, CTL Europe, Germany). Ten microlitres of the stained cell suspension were pipetted into the counting chamber and cell counting was performed on an ImmunoSpot Ultimate UV Image analyzer (CTL Europe, Germany). PBMCs were either directly used in multiparameter Fluorospot assays and flow cytometry-based ICS or $5 \times 10^6$ PBMCs were cryopreserved per vial in 1.8 mL cryotubes (Thermo Scientific, Denmark) at a concentration of $1 \times 10^7$ PBMC per 1 mL freezing medium (fetal calf serum (FCS) (Life Technologies, Germany), supplemented with 10% DMSO (Sigma-Aldrich, Germany), using a freezing container (Thermo Scientific, Denmark) and stored at $-80\,°C$. After 24 h, PBMCs were stored in the vapor phase of a liquid nitrogen tank until further use.

**Multi-parameter IFNγ/IL-2/TNF/Granzyme B (GzmB) and IL-5/IL-4 Fluorospot assays for detection of the frequencies of spike-reactive PBMCs directly ex vivo.** Human IFNγ/IL-2/TNF/GzmB four-color and human IL-5/IL-4 dual-color Fluorospot assays (CTL Europe, Germany) were performed according to the manufacturer's instructions for single parameter Fluorospot assays. One day before the Fluorospot assays were performed, the plates were activated by adding 70% ethanol for less than one minute, followed by a washing step and addition of IFNγ/IL-2/TNF/ GzmB or IL-5/IL-4 capture antibodies overnight, respectively. After decanting the plate, PBMCs were used directly after isolation and placed at $2 \times 10^5$ (4CFS) or $8 \times 10^5$/well (2CFS) in a final volume of 200 µL/well. PBMCs were then stimulated for 22 h (4CFS) or 48 h (2CFS) with 1 µg/mL of overlapping peptide pools (15mers overlapping by 11 aa) of the SARS-CoV-2 spike protein (PepMix™ SARS-CoV-2 (PM-WCPV-S), consisting of two peptide pools, i.e., S1 and S2 with 158 and 157 peptides, respectively) from JPT Peptide Technologies, Germany). The S1 peptide pool (158 15mer peptides) covered the N-terminal amino acid (aa) residues 1–643, and the S2 peptide pool (156 15mer peptides and one 17-mer at the C-terminus) encompassed the C-terminal aa residues 633–1273[4], and of the SARS-CoV-2 nucleoprotein (PepMix™ SARS-CoV-2, PM-WCPV-NCAP), consisting of 102 peptides. As an antigen-specific positive control, we used a CEF pool of in total thirty-two 15mer peptides derived from Cytomegalovirus (5 peptides), Epstein-Barr virus (15 peptides), and Influenza virus (Flu) (12 peptides) proteins (National Institute for Biological Standards and Control, UK).

After the stimulation period, the plates were washed and 80 µL of either anti-human IFNγ (FITC), anti-human IL-2 (Hapten2), anti-human TNF (Hapten1), and anti-human GzmB (Biotin) or anti-human IL-5 (Hapten3) and anti-human IL-4 (Biotin) detection antibody solution was added for additionally 2 h. For the visualization of secreted cytokines and GzmB, plates were washed and a tertiary solution including either anti-FITC Alexa Fluor® 488 (visualizes IFNγ), anti-Hapten2 CTL-Red™ (visualizes IL-2), anti-Hapten1 CTLYellow™ (visualizes TNF), and Streptavidin eFluor® 450 (visualizes GzmB) or anti-Hapten3 Alexa Fluor® 488 (visualizes IL-5) and Strep CTL-Red™ (visualizes IL-4) was added for 1 h. The staining procedure was stopped by washing the plate.

After drying the plates for 24 h on paper towels, multi-parameter Fluorospot plates were scanned using an automated reader system (ImmunoSpot Ultimate UV Image analyzer/ImmunoSpot 7.0.17.0 Professional DC Software, CTL Europe GmbH, Germany). Counting of spot forming cells (SFC) on Fluorospot plates was performed by adjusting the sensitivity, background balance, and the gates for the spot size using the CTL software. Counting was performed in compliance with the guidelines for the automated evaluation of ELISpot assays. All counts were reviewed and certified by a second person in a rigorous quality control process. The final results are represented as SFCs per $1 \times 10^6$ PBMCs. Positive reactivity to experimental stimulatory agents was given when the spot count in antigen-stimulated cells was greater than twice the spot count in unstimulated (background) wells.

**Direct ex vivo detection of SARS-CoV-2 reactive T cells by flow cytometry using ICS.** Totally, $1 \times 10^6$ freshly isolated PBMCs were transferred into 150 µL RPMI1640 medium supplemented with 10% FCS and 1% penicillin-streptomycin (PenStrep, Life Technologies, Invitrogen, Germany) (abbr.: RPMI-10) containing costimulatory antibodies to ensure effective T cell stimulation (1 µg/mL anti-CD28; BD Biosciences, Germany) in one well of a 96-well polypropylene U-bottom microtiter plate. Cells were stimulated with the S1/S2 peptide pools (1 µg/mL) as mentioned above. After one hour of incubation at 37 °C in 5% CO2, 10 µg/mL of Brefeldin A (Sigma-Aldrich, Germany) was added to the cell suspension and incubated for 4 h at 37 °C in 5% CO2, after which ICS was performed.

Stimulated PBMCs were labeled with the LIVE/DEAD™ Fixable Blue Dead Cell Stain Kit (Thermo Fisher Scientific, USA) in a total volume of 100 µL for 30 min at 4 °C in the dark, and washed twice with 200 µL FACS buffer. After centrifugation (560g, 4 °C, 5 min), PBMCs were fixed for 20 min at 4 °C in the dark in 100 µL of an intracellular fixation buffer (Intracellular Fixation Buffer, Thermo Fisher Scientific, USA). After two wash steps with 200 µL/well Perm/Wash solution (Cytofix/Cytoperm Kit; BD Biosciences) and a centrifugation step (710 g, 4 °C, 5 min), PBMCs were stained with mouse anti-human CD28 (1.0 µg/mL, BD Biosciences, Cat# 555725), mouse anti-human CD28 (1.0 µg/mL, BD Biosciences, Cat# 555725).

BV510 mouse anti-human CD3 (1.0 µg/mL, BioLegend, Cat# 344828), EF450 mouse anti-human CD4 (0.5 µg/mL, eBioscience, Thermo Scientific, Cat# 48-0047-42), ECD mouse anti-human CD8 (0.2 µg/mL, Beckman Coulter), Al700 mouse anti-human IFNγ (0.1 µg/mL, BD Bioscience, Cat#557995), BV785 mouse anti-human TNF (5.0 µg/mL, BioLegend, Cat# 502948), FITC rat anti-human IL-2 (3.1 µg/mL, eBioscience, Thermo ScientificCat# 11-7029-42) in a total volume of 80 µL Perm/Wash buffer including a brilliant violet buffer (BD Pharmingen Stain Buffer, BD Biosciences) for 30 min at 4 °C in the dark. Single color compensation was performed using 25 µL of compensation beads (UltraComp eBeads and ArC™ Amine Reactive Compensation Bead Kit for LIVE/DEAD compensation, both from Thermo Fisher Scientific, USA) following the instructions of the manufacturer. Cells and beads were washed twice and finally re-suspended in 300 µL FACS buffer for acquisition. Cells were stored cold and in the dark until acquisition.

Acquisition of samples was performed within four hours after staining using an LSR2/LSR Fortessa flow cytometer and FACSDiva Software V.6.1.3 (Becton Dickinson, Germany). Photomultiplier voltages were adjusted with the help of unstained cells for all parameters. Analysis was performed on at least $1.5 \times 10^5$ living lymphocytes using the software FlowJo version 10.7.0 (FlowJo LLC, USA).

The gating strategy for flow cytometric analysis of ex vivo re-stimulated PBMCs is shown as Supplementary Fig. 9. Each gate was set in the negative control sample and then adjusted to peptide-stimulated cells with consideration of T cell receptor downregulation. Two independent audits by different persons were performed to control the gating. According to the differential expression of CD4 and CD8 T cell, subpopulations were defined.

After background subtraction, CD4 and CD8 T cell responses were calculated by summing up all events staining positive for expression of the cytokines IFNγ, TNF, or IL-2 and their respective combinations. Raw data of all assays are available upon request.

**Statistical analysis.** All results were included in the analysis, and no attempt was made to exclude outliers. All tests were two-sided and conducted on exploratory 5% significance levels. Nonparametric statistical tests were applied in all cases. Unpaired Mann-Whitney tests were performed to compare distributions of relevant variables between different groups of study individuals. Wilcoxon signed-rank testing was employed to assess the significance of the change in values among different time points. One-way ANOVA testing was applied for the comparison of different time points within the study groups. In the case of resulting $p$-values < 0.05, Mann–Whitney and Wilcoxon signed-rank tests were performed to assess the significance of the change in values between specific time points. Spearman correlation analysis was performed to evaluate the correlation of e.g., S1- or S2-reactive cytokine-secreting cells from convalescents with serum anti-SARS-CoV-2 IgG levels or with cell culture virus-neutralization activity. The software Graph Pad Prism 9.1.0 (GraphPad Software, La Jolla, CA, USA) was used for statistical analyses.

**Reporting summary.** Further information on research design is available in the Nature Research Reporting Summary linked to this article.

## Data availability

All data generated in this paper are provided in the Supplementary Information/Source Data Files. The study protocols for the SeCoMRI and VaCoMRI studies are available upon request. Source data are provided with this paper.

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

## Acknowledgements
The study was funded by the Network of German University Hospitals (NUM to P.A.K., U.P., and P.L.), the German Center for Infection Research Munich site, the FOR-COVID initiative of the state of Bavaria (P.A.K. and U.P.), the project "Virological and immu-nological determinants of COVID-19 pathogenesis—lessons to get prepared for future pandemics (KA1-Co-02 "COVIPA"), a grant from the Helmholtz Association's Initiative and Networking Fund (P.A.K. and U.P.), the SFB TRR 179 (P.A.K. and U.P.) and faculty resources (P.A.K., U.P., and P.L.), the CoVRapid Grant from the Bayerische For-schungsstiftung (P.A.K. and U.P.), and the ForCovID Grant of the State of Bavaria (U.P., O.P., and P.A.K.).

## Author contributions
N.K., A.P., S.Y., T.B., H.M., C.C.C., H.W., S.J., M.G., M.F., E.V., and B.H. performed the experiments and evaluated the data. A.P., S.Y., O.Z., H.R., M.H., C.C., J.E., K.T., P.L., O.K., D.Z., U.P., and P.A.K. developed the study design and conducted the study; C.W. and J.R. provided the essential reagents; U.P. and P.A.K. wrote the paper, all authors approved the paper.

## Funding

## Competing interests
The authors declare no competing interests.
