## [Peer Review File · Nature Communications]

Dynamics of spike- and nucleocapsid specific immunity during long-term follow-up and vaccination of SARS-CoV-2 convalescentsEditorial Note: This manuscript has been previously reviewed at another journal that is not operating a transparent peer review scheme. This document only contains reviewer comments and rebuttal letters for versions considered at Nature Communications.

Reviewers' Comments:

Reviewer #2:

Remarks to the Author:

As pointed out in my previous comments the work entitled "Long-term follow-up of SARS-CoV-2 convalescents reveals distinct magnitude of spike-specific immunity after viral re-exposure and vaccination" is a technically well executed work that mainly confirmed numerous reports showing that antibodies declined more rapidly than T cells in convalescent individuals and that vaccination in convalescent individuals elicit more rapidly a Spike-specific antibodies and T cell response. The analysis of such T cell response is technically well executed, even though the novelty of the "IL-2" analysis suggested by the authors is not real since other papers (I.e. Differential effects of the second SARS-CoV-2 mRNA vaccine dose on T cell immunity in naïve and COVID-19 recovered individuals. Cell Reports 2021) have also analyzed IL-2 related T cell response in these groups. In addition the authors are stating that they "Fluorospot assay is an high sensitive assay . However there are no data that show that such assay is more sensitive than other assays currently used by others groups directly ex vivo . It might be more informative in term of analysis of multifunctionality but the term "highly sensitive" is speculative. The authors should demonstrate that such assay (likely done in frozen separated PBMC) is more sensitive than other assays performed for example in fresh blood (and directly measuring different cytokines quantity).

Major Points

1-As pointed out in my previous comment, the "novelty" of the work is mainly restricted to the studies of very few individuals (6 convalescent individuals) which based on the surge of Spike-specific antibodies are likely to have been in contact with the virus. I agree with the authors that "The most likely explanation for the sudden increase in anti-SARS-CoV-2 antibodies after a relatively long time-span is therefore re-exposure to SARS-CoV-2. " Nevertheless, this remains an hypothesis and such re-exposure might not results in an full re-infection (it can be just an abortive infection) and it might be completely different in subjects with real breakthrough infection.. As such using the data from these selected individuals to conclude that "our results show distinct magnitudes of spike-specific immunity in convalescents after viral re-exposure compared to vaccination" remains highly speculative. In reality, I don't think that the authors characterized "the immune response of convalescents from mild SARS-CoV-2 infection, ..., react to antigen re-encounter after re-exposure to SARS-CoV-2 compared to COVID-19 vaccination." As claimed in the introduction (line 94-96)

To demonstrate that Spike-specific T cell response in convalescent might be of different magnitude after "viral re-exposure" than vaccination, the authors should study not only convalescent with a surge of antibodies but also convalescent individuals with a demonstrated breakthrough infection.

2) The authors use the term "virus-specific polyfunctional T cells" to indicate their results of T cells specific for Spike and NP. As I pointed out before, SARS-CoV-2 T cell response is not only directed towards Spike (and NP) . The authors have performed now a further analysis of NP-specific T cells, which represents a good addition. Still, they cannot conclude, that the absence of Spike and NP specific T cells demonstrated a global absence of circulating SARS-CoV-2-specific T cells in the "exposed individuals". NP and Spike are only a minor part of the whole SARS-CoV-2 proteome and

several works have actually show that T cell response against ORF-1 coded non structural proteins often represent the dominant T cell response(see Ferretti et al Unbiased Screens Show CD8+ T Cells of COVID-19 Patients Recognize Shared Epitopes in SARS-CoV-2 that Largely Reside outside the Spike Protein. Immunity 2020). Thus, the discussion related to "tissue resident T and B cells" is interesting but remains speculative and the limitation of testing only a minority of the possible SARS-CoV-2 T cell repertoire should at least be also discussed.

Point-by-point response to Reviewer 2

Reviewer #2 (Remarks to the Author):

As pointed out in my previous comments the work entitled “Long-term follow-up of SARS-CoV-2 convalescents reveals distinct magnitude of spike-specific immunity after viral re-exposure and vaccination” is a technically well executed work that mainly confirmed numerous reports showing that antibodies declined more rapidly than T cells in convalescent individuals and that vaccination in convalescent individuals elicit more rapidly a Spike-specific antibodies and T cell response. The analysis of such T cell response is technically well executed, even though the novelty of the “ IL-2” analysis suggested by the authors is not real since other papers (I.e. Differential effects of the second SARS-CoV-2 mRNA vaccine dose on T cell immunity in naïve and COVID-19 recovered individuals. Cell Reports 2021) have also analyzed IL-2 related T cell response in these groups. In addition the authors are stating that they “Fluorospot assay is an high sensitive assay. However there are no data that show that such assay is more sensitive than other assays currently used by others groups directly ex vivo. It might be more informative in term of analysis of multifunctionality but the term “highly sensitive” is speculative. The authors should demonstrate that such assay (likely done in frozen separated PBMC) is more sensitive than other assays performed for example in fresh blood (and directly measuring different cytokines quantity).

Answer: We thank the reviewer for the positive evaluation of our results. We would like to clarify that we did all analyses on fresh PBMC directly after isolation.

We have now cited the publication by Zolano et al in Cell Reports 2021 as suggested by the reviewer on page 12, where we describe the results of Figure 5a.

In addition, we have removed any wording that could create the impression that we would directly compare the Fluorospot to flow-cytometry based assays. It was never our purpose to describe the four color Fluorospot as the most sensitive assay in our manuscript. Therefore, all wording with potential for such a misunderstanding has been removed from pages 5 and 6 of the revised manuscript.

Major Points

1-As pointed out in my previous comment, the “novelty” of the work is mainly restricted to the studies of very few individuals (6 convalescent individuals) which based on the surge of Spike-specific antibodies are likely to have been in contact with the virus. I agree with the authors that “The most likely explanation for the sudden increase in anti-SARS- CoV-2 antibodies after a relatively long time-span is therefore re-exposure to SARS-CoV-2.” Nevertheless, this remains an hypothesis and such re-exposure might not results in an full re-infection (it can be just an

abortive infection) and it might be completely different in subjects with real breakthrough infection.. As such using the data from these selected individuals to conclude that “ our results show distinct magnitudes of spike-specific immunity in convalescents after viral re-exposure compared to vaccination” remains highly speculative. In reality, I don't think that the authors characterized “ the immune response of convalescents from mild SARS-CoV-2 infection, ..., react to antigen re-encounter after re-exposure to SARS-CoV-2 compared to COVID-19 vaccination.” As claimed in the introduction (line 94-96).

Answer: We have removed any claims from the manuscript (title, abstract, introduction, results, figure legends, extended data figure legends and discussion) that a re-exposure to SARS-CoV-2 explains the rise in virus-neutralizing antibody titers. As proposed by the reviewer, we now provide a balanced discussion on the possible causes of an increase in virus-neutralizing antibody titers without drawing any firm conclusions and added the possibility of an abortive infection.

To demonstrate that Spike-specific T cell response in convalescent might be of different magnitude after “ viral re-exposure” than vaccination, the authors should study not only convalescent with a surge of antibodies but also convalescent individuals with a demonstrated breakthrough infection.

Answer: As the reviewer encouraged us to include this aspect, we have included a careful discussion of breakthrough infections in the discussion. The situation we have studied during the second wave of the pandemic is clearly different from the current situation, where the highly infectious predominant SARS-CoV-2 delta variant is the predominant viral strain causing breakthrough infection in vaccinated but also convalescent individuals with waning immunity.

2) The authors use the term “virus-specific polyfunctional T cells” to indicate their results of T cells specific for Spike and NP. As I pointed out before, SARS-CoV-2 T cell response is not only directed towards Spike (and NP). The authors have performed now a further analysis of NP-specific T cells, which represents a good addition. Still, they cannot conclude, that the absence of Spike and NP specific T cells demonstrated a global absence of circulating SARS-CoV-2-specific T cells in the “exposed individuals”. NP and Spike are only a minor part of the whole SARS-CoV-2 proteome and several works have actually show that T cell response against ORF-1 coded non structural proteins often represent the dominant T cell response(see Ferretti et al Unbiased Screens Show CD8+ T Cells of COVID-19 Patients Recognize Shared Epitopes in SARS-CoV-2 that Largely Reside outside the Spike Protein. Immunity 2020). Thus, the discussion related to “tissue resident T and B cells” is interesting but remains speculative and the limitation of testing only a minority of the possible SARS-CoV-2 T cell repertoire should at least be also discussed.

Answer: We apologize for this misunderstanding. We introduced the term polyfunctional to describe T cells that produce at least 2 cytokines upon peptide-specific activation. We have explained the term polyfunctional now on page 6 of the manuscript to avoid any misunderstandings.

We have further followed the reviewer's suggestions and mentioned in the on page 16 in the results section the possibility that T cell responses against non-structural antigens of SARS-CoV-2 have remained undetected in our study and cited the publication suggested by the reviewer.